# Sampling and Analysis of Low-Molecular-Weight Volatile Metabolites in Cellular Headspace and Mouse Breath

**DOI:** 10.3390/metabo12070599

**Published:** 2022-06-27

**Authors:** Theo Issitt, Sean T. Sweeney, William J. Brackenbury, Kelly R. Redeker

**Affiliations:** 1Department of Biology, University of York, York YO10 5DD, UK; ti538@york.ac.uk (T.I.); sean.sweeney@york.ac.uk (S.T.S.); william.brackenbury@york.ac.uk (W.J.B.); 2York Biomedical Research Institute, University of York, York YO10 5DD, UK

**Keywords:** volatile organic compound, VOC, headspace, breath, breath biomarker, volatile metabolite, breath diagnosis

## Abstract

Volatile compounds, abundant in breath, can be used to accurately diagnose and monitor a range of medical conditions. This offers a noninvasive, low-cost approach with screening applications; however, the uptake of this diagnostic approach has been limited by conflicting published outcomes. Most published reports rely on large scale screening of the public, at single time points and without reference to ambient air. Here, we present a novel approach to volatile sampling from cellular headspace and mouse breath that incorporates multi-time-point analysis and ambient air subtraction revealing compound flux as an effective proxy of active metabolism. This approach to investigating breath volatiles offers a new avenue for disease biomarker discovery and diagnosis. Using gas chromatography mass spectrometry (GC/MS), we focus on low molecular weight, metabolic substrate/by-product compounds and demonstrate that this noninvasive technique is sensitive (reproducible at ~1 µg cellular protein, or ~500,000 cells) and capable of precisely determining cell type, status and treatment. Isolated cellular models represent components of larger mammalian systems, and we show that stress- and pathology-indicative compounds are detectable in mice, supporting further investigation using this methodology as a tool to identify volatile targets in human patients.

## 1. Introduction

Volatile Organic Compounds (VOCs) are small, carbon-containing compounds that are found, at least partially, in the gaseous state at standard temperature and pressure. The human ‘volatilome’ describes the VOCs that are produced and metabolised within the human body [1]. These compounds provide valuable insights into metabolic processes and can be detected from the breath, skin, urine, faeces and saliva [1,2], providing an opportunity to diagnose and monitor treatment as well as measure bodily functions.

A large amount of research has been conducted upon human breath with a range of VOCs linked to disease [3]. However, in the field of breath and ‘smell’ diagnostics, more human research (e.g., sampling individually and directly from breath) has thus far been conducted than research that tests volatile outcomes from preclinical, pathogenically representative cellular models, limiting the mechanistic understanding of the VOC metabolism. There is a paucity of published research linking cellular processes and VOC metabolisms to identify diagnostically powerful and translatable VOC biomarkers of cellular and disease processes [3]. We focus here upon breath, as it provides insights into systemic, internal bodily processes via the diffusion between the lungs and blood.

Many methodological approaches for breath VOC collection have been described [4,5,6,7], and some metabolic processes have been linked to the volatilome, such as the reactive oxygen species production of aldehydes and alkanes [1,8,9] and microbial function linked to sulphur compounds such as dimethyl sulphide [3,10]. Diagnostic applications of VOCs remain limited in the clinic, in part due to conflicting and confounding results [3].

Useful VOC biomarkers should be descriptive of a condition while overcoming environmental, individual and methodological variabilities. Reported breath VOC variability accrues from individual comorbidities and variations in analytical and collection methods, leading to reduced sensitivity and a lack of recognition of potentially useful biomarker compounds. Commonly used methodological approaches also rely on single time-point sampling and do not take into account the ambient volatile environment, allowing environmental variability to influence and reduce reported outcome precision [3,11,12], relying instead upon substantial deviations from the norm and reducing the utility of breath volatiles through a loss of information. New approaches and perspectives are therefore needed to contextualise the valuable research done so far and to identify robust volatile biomarkers to provide fast, noninvasive and low-cost diagnostics.

The metabolism of VOCs, defined here as flux (reported in grams compound per gram organism weight per time, i.e., g g^−1^ s^−1^), considers both release and consumption. The production of compounds can be an expression of metabolic products, for example, acetone release in the breath from alterations in the metabolism [13] and carbon dioxide release from the glucose metabolism. The emissions of VOCs may also be caused by the release from saturated tissues, such as the muscular release of isoprene in human breath during exercise [14]. The consumption of VOCs can also be observed through the active metabolism, such as with CYP450 enzymes in the human liver [15] or the consumption of oxygen. Quantifying and understanding a healthy human metabolism and its impact on VOCs is a developing field and is necessary to define population variability and compound-specific standard ranges in human breath.

The uptake of compounds is not reported as often as its release [3], and so volatile ‘sinks’, the use of VOCs by cells as substrates, may be overlooked as a result of collection methodology and analytical focus using nontargeted gas chromatography mass spectrometry techniques. Nontargeted approaches primarily detect relatively concentrated material (ppbv), whereas targeted approaches are generally capable of quantifying at much lower concentrations (pptv).

In the case of disease, understanding systemic uptake/release is critical in the development of biomarkers for clinical application. Disease metabolism outcomes depend upon compound reactivity, transportation time spent within active metabolic regions or saturated tissues and active metabolic by-products and interactions with the disease pathology [3]. Alterations in the VOC flux stem from cellular environmental changes which influence metabolic response, either as a result of dysfunction or as the result of normal processes, such as exercise. The identification and separation of these processes in the volatilome is challenging because many cellular processes, dysfunctional or otherwise, produce similar changes in the environmental and physiological state. For example, a shift towards glycolysis in cancer [16,17], or mitochondrial dysfunction [18], may result in similar global/tissue alterations in pH and reactive oxygen species, producing VOCs associated with this change. Breath volatiles can also be seen to change as a result of normal metabolisms, such as with fasting and eating [19,20] or with circadian rhythms [21]. It is therefore important to be able to identify and characterise variation in the cellular type, status (disease) and response to environmental stress.

There exists an ever-increasing body of work using cellular and preclinical models for biomarker discovery [3,22]. Some pertinent examples include the volatile emissions of cancer cells, such as hepatic [23], gastric [24], breast [25] and lung [26], along with noncancerous cells such as stem cells [27], endothelial [28] and human fibroblasts [26]. Nonhuman pathogens have also been investigated as potential avenues for breath biomarker discovery, including infectious lung agents such as Aspergillus [29]. The majority of studies investigate volatile emissions (by definition, the production or discharge of a compound from an organism or system), generally through the comparison of single time-point sample headspace peak area/concentrations relative to a background or control. These studies identify compounds which are produced by an organism/cell, but they do not quantify the rate at which the metabolism occurs. Uptake by human cells has been observed [22,23]; however, compound consumption metabolisms are neither quantified nor understood, and longitudinal studies of cellular/organism metabolisms is a developing area of research. Here, we present a novel approach utilizing multiple time points to allow the quantification of a volatile compound metabolism (where metabolism, or flux, incorporates the potential for either emission or consumption) and which allows for longitudinal approaches that can identify and quantify changes in the active cellular processing of volatiles.

This study aims to characterise the volatile metabolisms of a select suite of volatiles in different cell types and statuses (disease). We hypothesise that volatiles collected from the headspace above these cells will differ significantly. Furthermore, treatment with a chemotherapeutic agent, Doxorubicin, will also produce significant, detectable alterations in the volatiles metabolised by the cells. These volatile metabolisms, linked to phenotype and pathophysiology, could provide potential targets for diagnostic research. To test the translatability of the method and select volatiles, mice breath will also be investigated.

## 2. Results

### 2.1. Volatile Flux in Cellular Headspace

The methodological approach is outlined in Figure 1a. Headspace sampling from custom chambers (Figure 1b) from multiple time points allows calculation of cellular volatile fluxes (pg/ug/h^−1^).

Headspace analysis was conducted for media only and all supplementation (dimethyl sulfoxide (DMSO) and doxorubicin) controls (Figure 1 and Appendix A). No significant variation was observed between Dulbecco’s Modified Eagle’s Medium (DMEM), DMEM:F12 media (Appendix A) or with the addition of DMSO (Appendix A). Because no variation was observed between DMEM and DMEM:F12 with the DMSO addition, DMSO values represent a combination of DMEM (*n* = 3) and DMEM:F12 (*n* = 3) with the DMSO addition.

Headspace above cells had appropriate media controls (average) deducted, demonstrated in Figure 1C and Appendix A. This was then normalised to protein content (Figure 1D and Appendix A) to give the ug of the compound per hour per ug of protein. This is shown for MDA-MB-231 cells, but the media subtraction process was repeated for each cell line and treatment.

### 2.2. Volatile Profiles by Cell Type

Comparison of cells growing at basal capacity (i.e., in fully supplemented, optimum media) within a laboratory setting revealed differences in selected volatiles in the headspace. Methyl chloride (MeCl), isoprene and acetone significantly differ between cell lines. Cancer cell lines show consistently higher levels of MeCl and acetone compared to noncancer cell lines.

#### 2.2.1. Headspace Volatiles Differ between Breast and Kidney Derived Cells

For noncancer cells (Figure 2A), HEK293t cells show significant uptake of MeCl compared to MCF10A and a significant release of isoprene (Figure 2A). HEK293T cells consumed significantly more acetone than MCF10a, and M57 uptake was also increased (Figure 2A). In contrast, 2-methyl pentane (2-MP) production appears increased in HEK293T cells compared to MCF10A (Figure 2A).

#### 2.2.2. Headspace Volatiles Differ between Cancer and Noncancer Breast Epithelial Cells

When comparing the headspace samples from breast cancer MCF7 and MDA-MB-231 to those of noncancer MCF10A cells derived from breast tissue (Figure 2A–B), MeCl levels were enhanced over MCF7 and were significantly enhanced over MDA-MB-231 cells compared with MCF10A. Methyl bromide (MeBr) and dimethyl sulphide (DMS) levels were increased over MDA-MB-231 cells compared to both MCF7 and MCF10A. MCF7 cells exhibited significantly increased production of isoprene compared to MCF10A, which exhibited isoprene uptake. MDA-MB-231 cells also revealed the production of isoprene rather than consumption. Acetone uptake is reduced in MCF7 cells compared to MCF10A and MDA-MD-231 and show significant changes in the production of acetone; however, the range is large (Figure 2B). M57 was increased in MDA-MB-231 cells compared with MCF10A.

#### 2.2.3. Headspace Volatiles Differ between Cancer and Noncancer Kidney-Derived Cells

For cells derived from kidney (Figure 2C), HEK293T cells showed the uptake of MeCl, which is unique when compared to all other untreated cells lines. RCC4 cells showed little production or consumption of MeCl. Isoprene was significantly more concentrated in the headspace of HEK293T cells compared to RCC4, which showed a metabolic uptake. Acetone consumption was significantly reduced in RCC4 cells compared to HEK293t (Figure 2C). RCC4 cells showed some uptake of 2-MP vs. HEK293T production, with increased production of n-hexane vs. HEK293T (Figure 2C).

### 2.3. Effects of Chemotherapeutic Agent, Doxorubicin, upon Cellular Volatile Profiles

Doxorubicin treatment produced significant alterations in the volatile profile of both MCF10A and MDA-MB-231 cells, as shown in Figure 3. The treatment of MDA-MB-231 with 250 nM and 750 nM revealed consistent trends with increasing concentrations (Appendix A). For the MDA-MB-231 cells, MeCl switched significantly from production to uptake with increasing concentrations of doxorubicin. Methanethiol (MeSH) also showed increased uptake, while DMS was significantly increased in its release. The uptake of acetone by the MDA-MB-231 cells was observed, but it was nonsignificant. Significant uptake by MDA-MB-231 cells was observed for M57, with no change in MCF10A. Doxorubicin also produced significant increases in 3-methyl pentane (3-MP) and provoked n-hexane release in MDA-MB-231 cells. MCF10A cell volatiles changed in a similar manner as MDA-MB-231 in response to the doxorubicin treatment. MeCl showed a similar shift to uptake from production, where DMS production was increased, and chloroform (CHCl3) was produced.

MTT assay was performed as an indication of metabolic activity. MCF10A cells show greater metabolic activity than the MDA-MB-231 cells. Treatment with doxorubicin increased the metabolic activity by this assay compared to vehicle (Appendix A). The sulphorhodamine B (SRB) assay revealed no significant variations for cell growth at 24 h between treatments. At 48 h, the doxorubicin treatment suppressed growth in both cell lines (Appendix A). Trypan blue exclusion revealed a nonsignificant reduction in cell viability at 370 and 740 nM doxorubicin for MDA-MB-231 cells and a similar but significant reduction in cell viability in MCF10a cells exposed to 740 nM doxorubicin (Appendix A).

### 2.4. Breath and Faecal Volatiles from Mice

Collection of breath from 9-week-old female Rag2^−/−^ Il2rg^−/−^ mice using the sampling chambers (Figure 1B) reveals metabolic interaction with several volatile compounds (Figure 4). Because the mice were allowed to behave normally in the chambers for 20 min following 10 min of acclimatisation, the presence of mouse (in white, Figure 4) is representative of both mouse breath and faecal volatiles, whereas faecal (in orange, Figure 4) indicates faecal material volatiles only.

Mice show significant positive production of MeCl compared to faecal material, as well as the production of isoprene (Figure 4A). The 3-MP uptake by mice is significant, although the uptake is reduced by the presence of faecal matter (which generally produced 3-MP) (Figure 4C).

## 3. Discussion

This research demonstrates that volatile analysis is capable of separating cellular models by cellular type, disease status and response to chemically induced stress. Furthermore, we have shown that representative, discrete indicator compounds are found in mouse breath and are both actively produced and metabolised. A selection of these compounds, including methyl halides, have recently been reported in human breath [30]. These outcomes support further research into their potential use as biomarkers of disease.

We have quantified volatile signatures (12 discrete compounds via SIM) of cells derived from two tissues and disease pathologies and revealed how environmental and cellular changes elicit detectable alterations in the healthy cell volatilome through treatment with the chemotherapy drug Doxorubicin. These volatile metabolisms, linked to phenotype and pathophysiology, provide potential targets for diagnostic research. We have demonstrated how these cellular models are applicable in mammalian analysis through quantification of mice breath volatiles, targeting the specific compounds which have shown most promise in these early analyses.

These analyses rely upon a novel, noninvasive volatile sampling method, which allows the multi-time-point analysis of VOC concentrations in cellular headspace, and which can be used in an ethically appropriate manner with mice volatile sampling. This multi-time-point approach allows a more comprehensive understanding of cellular/mice/faecal volatile fluxes, including both emissions and consumption. Further to this end, we use both targeted mass spectrometry, or ‘selective-ion mode’ (SIM), to maximise sensitivity and reproducibility, along with SCAN modes to identify new biomarker targets.

### 3.1. Cellular Volatiles and Metabolisms

Compared to the number of human breath studies, there are limited studies investigating cellular headspace volatile concentrations, and less on volatile metabolisms. This is an important avenue of research for breath biomarker discovery. Headspace volatiles for MCF10A, MCF7 and MDA-MB-231 cells have previously been investigated [9,25,31]. HEK293T cells have also had some limited investigation [32]. This is the first time that RCC4 cell headspace volatiles have been reported. In this work, we have focused on a novel approach to describe the dynamics of 12 selected VOCs, reflective of cellular metabolisms, not the discovery of new volatiles using nontargeting approaches. This allows for greater precision and resolution in the assessment of select VOC dynamics, which is well suited to a longitudinal approach. Of these 12 VOCs, this is the first report of methyl halide and chloroform metabolism in human cells and the breath of mice.

A challenge in volatile breath research is the paucity of data regarding metabolic processes and alterations dependent upon compound and/or cellular type/state. For example, while chloroform exposure is well documented, and the compound is broken down in the liver by CYP450 enzymes [33], its (normal) metabolic consumption and production in mammalian systems has not previously been described.

Likewise, human erythrocytes contain a glutathione-s-transferase isoenzyme that metabolises methyl halides [34,35], but this is not present in all humans [36]. Methyl halide metabolism remains unidentified and undescribed in human systems. All plants and fungi measured to date produce methyl halides, but the functional reason for this metabolism remains unclear [37]. A role for active metabolism of methyl halides in mammalian systems is presented in this paper, as we have shown the active production and consumption of MeCl, MeBr and MeI in varying situations. Their potential as disease biomarkers, however, requires further research.

In our tested cellular systems, the metabolism of MeCl is descriptive of the cellular type, with cancer cells exhibiting increased release relative to their healthy controls. Under the treatment of Doxorubicin, MeCl uptake is seen in response. Furthermore, this compound can be quantified in the breath of mice and humans. The association of methyl halides with mammalian systems has been limited, and overexposure of MeCl in rats was not linked to DNA adducts, where MeI and MeBr have been shown to cause systemic DNA methylation [38]. Long-term exposure of MeCl at high concentrations (1000 ppm) produced renal tumours in male rats and glutathione depletion [39,40].

MeSH and DMS are linked as sulphur-containing compounds and are metabolites for each other, with MeSH serving as a precursor to DMS (with a methylating agent) and DMS serving as a precursor to MeSH (with a demethylating agent) [41,42]. The glutathione (GSH)-based metabolism of MeCl can result in the formation of MeSH [43]. Both MeSH and DMS have been linked to bacterial processing [10,44]. HepG2 (hepatocarcinoma cells) and TBE (tracheobronchial epithelial cells) have been shown to produce DMS [23,45], whereas we have only shown the production in MDA-MB-231 cells and in MCF10a and MDA-MB-231 cells following treatment with Doxorubicin. Sulphur-containing VOCs have been shown in human breath for a variety of diseases and processes [3]. Sulphur is also a dietary requirement [45], which suggests that diet will impact sulphur-volatile metabolism, and breath volatile concentrations, in individuals.

Isoprene and isoprenoids, as endogenous biomarkers, have been shown to be linked in patients with muscular dystrophy and are outputs of the mevalonate pathway [46]. Monitoring their levels may be important in a variety of diseases, such as cancer, as isoprenoids have been shown to be important compounds in tumour biology [47]. However, large variability between individuals, as demonstrated here and in a recent review [3], show that this volatile, while the most abundant VOC in human breath, is a challenging biomarker for individual/cohort diagnoses. Longitudinal and metabolic approaches, such as those described here, may prove able to utilise biomarkers with high variability between individuals, but further research is required. Isoprene has been reported to be utilised by HepG2 cells [23]. Here, we have shown a clear isoprene production by HEK293t and the uptake by MCF10a and RCC4.

Alkanes have been associated with oxidative stress and reactive oxygen species induced lipid peroxidation, linked to a range of diseases [48]. Moreover, 2- and 3-methyl pentane have been identified as potential markers of cancer [49,50], as has hexane [51]. Additionally, 2-MP has been shown to be produced by the lung cancer cell line NCI-H2807 [52], whereas we have only shown the production by HEK293t cells. Furthermore, 3-MP uptake has been demonstrated in the lung cancer cell line A549 [53,54], while we have only shown a significant production in response to Doxorubicin treatment in MDA-MB-231 cells. Alkanes are found in the breath of patients with a range of diseases, but it is prevalent in cancer [3]. Methylated alkanes are also descriptive of oxidative stress in transplant rejection [1,55]. However, the interplay between methylated and straight chain alkanes is less understood, and so six carbon alkanes were targeted here. MCF-7 cells have been shown to release alkanes in response to oxidative stress [9], which is supported by the release of 3-MP and hexane in response to Doxorubicin, which has also been shown to induce oxidative stress [56,57].

Of the compounds reported here, acetone is one of the most well-documented, and has been identified as a volatile compound associated with altered metabolisms and the development of ketosis [13]. Therefore, its dynamics are of interest in models of cancer which show altered energy processing. The uptake of acetone has been shown in the headspace of A549 and TBE cells [45,54], but emissions have been shown by VGP (vertical growth phase melanoma cells) [58] and A549 cells [26]. We have not shown consistent acetone production in any cell lines here, but varying levels of consumption across all cells. HEK293t cells consumed the most acetone and cancerous cells showed relatively less consumption against noncancerous cells.

We have shown both novel VOC targets and targets previously identified in cellular headspace and breath. We propose that the characterization of volatiles relative to cell type and status will allow for the utilization of a “breath-print” approach, where multiple volatiles indicative of specific healthy states or pathologies are combined to provide accurate and specific disease indicators. The refinement of target VOCs will increase with further research, and we have recommended research frameworks previously [3].

### 3.2. Mouse Volatiles

Our approach minimises stress in animals, which directly influences the breathing profile [59,60]. This longitudinal approach also allows us to view the compounds which are being metabolised/absorbed by mice and/or their faecal matter. As with humans, these mice show the release of MeCl; however, it is of note that these mice are immunocompromised, and their breath volatiles may differ from the standard wild-type mice models.

The identified active metabolisms of VOCs in mice provide targets for future disease mouse models and translates well into the breath of humans [30]. Here, we show variability over time and individual variability in mouse breath. With further research, the expected and average human range for each compound may be understood so to produce standards for medical application. However, individual variability over time supports a longitudinal approach to diagnosis, as direct comparisons between individuals may confound results.

## 4. Materials and Methods

### 4.1. Cell Culture and Treatment Conditions

Breast cancer cell lines MDA-MB-231 and MCF7 and kidney-derived cell lines; HEK-293t and RCC4 were grown in Dulbecco’s Modified Eagle Medium (DMEM, Thermo Scientific, Waltham, MA, USA), 25 mM glucose, supplemented with L-glutamine (4 mM) and 5% foetal bovine serum (Thermo Scientific, Waltham, MA, USA). The nontransformed human epithelial mammary cell line MCF10A was grown in DMEM/F12 (Thermo Scientific, Waltham, MA, USA) supplemented with 5% FBS, 4 mM L-glutamine (Thermo Scientific, Waltham, MA, USA), 20 ng/mL EGF (Sigma-Aldrich, Roche; Mannheim, Germany), 0.5 mg/mL hydrocortisone (Sigma-Aldrich, Burlington, MA, USA), 100 ng/mL cholera toxin (Sigma-Aldrich, Burlington, MA, USA) and 10 µg/mL insulin (Sigma-Aldrich, Burlington, MA, USA). All cell culture media was supplemented with 0.1 mM NaI and 1 mM NaBr (to model physiological availability of iodine and bromide). All cells were grown at 37 °C with 5% CO_2_.

MDA-MB-231 and MCF7 cells were a gift from Dr Mustafa Djamgoz. MCF10A were a gift from Dr. Norman Maitland, while HEK293t were a gift from Dr. Jared Cartwright and RCC4 were a gift from Dr. Dimitris Lagos.

To initiate the volatile collection, the procedure cells were trypsinised, and ~500,000 cells were seeded into 8 mL complete media. Cells were then allowed to attach for 3 h, washed with warm PBS 2× and an 8 mL treatment media was applied. Volatile headspace sampling was performed 24 h later.

Doxorubicin was dissolved in DMSO. Doxorubicin treatment was applied in DMEM 25 mM glucose, supplemented with L-glutamine (4 mM) and 5% FBS for the MDA-MB-231 cells and treatment medium for MCF10A. Appropriate doxorubicin concentration was determined using MTT and SRB assays, which assess metabolic activity and protein concentration as a measure of growth, respectively. Concentrations for doxorubicin treatment were chosen based on no less than 25% reduction in growth of metabolic activity following 24 h of treatment and supporting evidence in the literature of similar concentrations, eliciting senescent and maintaining growth [56,57,61]. This was determined by SRB, MTT and trypan blue exclusion assays (Appendix A). An amount of 750 nM was chosen to induce chronic cell stress over this time period while reducing the amount of cell death.

### 4.2. Headspace and Breath Sampling

#### 4.2.1. Cellular Headspace Sampling

Following the incubation period (24 h), 5 mL of supernatant medium was removed and plates, with lids removed, were placed into specially constructed chambers (Figure 1B) on a platform rocker on its slowest setting. Medium was equilibrated with lab air by flushing the chamber for 20 min using a Yamitsu air pump with a flow rate of 750 mL per min. Time zero (T0) samples were taken using an evacuated 500 mL electropolished stainless steel canister (LabCommerce, San Jose, USA) through Ascarite^®^ and Drierite^®^ traps [59]. The chamber headspace was then isolated by closing the lid valves and the chamber itself was left on the rocker for 120 min, at which point another air sample (T1) was collected. Cells were removed from the chamber, washed with PBS twice and lysed in 500 µL RIPA buffer (NaCl (5 M), 5 mL Tris-HCl (1 M, pH 8.0), 1 mL Nonidet P-40, 5 mL sodium deoxycholate (10 %), 1 mL SDS (10%)) with protease inhibitor (Sigma-Aldrich, Roche; Mannheim, Germany). Protein concentration of lysates were determined using the Bradford assay [62]. Background (medium only) readings were taken for all medium types and treatments, cell free and DMSO (vehicle), following 24 h incubation at 37 °C and 5% CO_2_ (Appendix A). DMSO concentration was used equivalent to the highest equivalent dose of doxorubicin; 0.000008%. These readings had no significant differences (determined by ANOVA) and were therefore pooled and the averages subtracted from each individual cell reading.

#### 4.2.2. Mouse Headspace Sampling

Nine-week-old female Rag2^−/−^ Il2rg^−/−^ mice were selected for sampling. This mouse strain is an immunocompromised model. Experimental replicates were 2 mice from a cage across 3 separate litters/cages: 6 mice in total. Experiments have been reported in-line with the ARRIVE guidelines.

Using tube handling methods, mice were gently placed with a cardboard tube and blue paper into the custom chambers. Flushing the chamber for 10 min using a Yamitsu air pump with a flow rate of 750 mL per min in undisturbed conditions, mice were allowed to acclimatise. T0 samples were then taken, and as with cellular headspace, the chambers were sealed for 20 min and T1 samples were then taken.

### 4.3. GC-MS, Calibration and Peak Analysis

Collected canister samples were transferred to a liquid nitrogen trap through a pressure differential. Pressure change between beginning and end of “injection” was measured, allowing calculation of the moles of gas injected. Sample in the trap was then transferred, via heated helium flow, to a Restek© (Bellefonte, PN, USA) PoraBond Q column (25 m length, 0.32 mm ID, 0.5-µm diameter thickness) connected to a quadrupole mass spectrometer (Aglient/HP 5972 MSD, Santa Clara, CA, USA). All samples here were analysed with a select ion mode (SIM) targeting the selected compound’s greatest detected mass unit. All samples were run within 6 days of collection. The oven program was as follows: 35 °C for 2 min, 10 °C/min to 115 °C, 1 °C/min to 131 °C and 25 °C min to 250 °C with a 5 min 30 sec hold. The quadrupole, ion source and transfer line temperatures were 280, 280 and 250 °C, respectively.

Calibration was performed using standard gases (BOC Specialty Gases, Woking, UK) and injections of various volumes, equal to different total amounts of compound. Linear regression analyses of calibration curves confirmed strong linear relationships between the observed SIM peak areas and moles of gas injected for each VOC (r2 > 0.9 in all cases). For compounds not purchased as speciality gases with ppbv concentration, 1–2 mL of compound in liquid phase was injected into a butyl sealed Wheaton-style glass vial (100 mL) and allowed to equilibrate for 1 h. An amount of 1 mL of headspace air was then removed using a gas tight syringe (Trajan, SGE) and injected into the headspace of a second 100 mL butyl sealed Wheaton-style glass vial. This was then repeated, and 1 mL of the 2nd serial dilution vial was injected into the GCMS system with 29 mL of lab air. This was performed for methanethiol (MeSH (SPEXorganics, St Neots, UK)), isoprene (Alfa Aesar, Ward Hill, MA, USA), acetone (Sigma-Aldrich, Burlington, MA, USA), 2- & 3-methyl pentane and n-hexane (Thermo Scientific, Waltham, MA, USA).

Nearly all reported compounds detected by the GC-MS were confirmed by matching retention times and mass–charge (*m*/*z*) ratios with known standards. This is in addition to a compound with retention time of 27.3, with masses 57 and 43 (M57), which, by relative distribution pattern, was determined, tentatively, to be 2-butanone from the NIST library and the human metabolome database [63].

Concentrations were calculated using peak area. Peak area/moles injected were calculated from previously generated calibration curves. Sample VOC concentrations were then normalised to CFC-11 concentrations (240 parts-per-trillion-by-volume (ppt)). CFC-11 was used as an internal standard, per sample standard for normalisation as atmospheric concentrations of CFC-11 are globally consistent and stable [64].

To account for differences in rates of proliferation (MCF10a cells proliferate at a higher rate than both MCF7 and 231 cells), results from GCMS analysis were normalised to protein content at time of sampling per plate using a Bradford assay [62].

### 4.4. Molecular Assays

#### 4.4.1. Sulphorhodamine B Assay

To determine cell growth, SRB assay was performed. The SRB assay measures cell density based on protein content [65]. Following incubation, cell monolayers were fixed with 10% (wt/vol) trichloroacetic acid (TCA) and stained for 30 min, after which the excess dye was removed by washing repeatedly with 1% (vol/vol) acetic acid. The protein-bound dye was dissolved in 10 mM Tris base solution for OD determination at 510 nm using a microplate reader [65].

#### 4.4.2. MTT Assay

MDA-MB-231 and MCF10A cells were seeded onto 96-well plates at a density of 8000 cells per well. Serial dilutions across the plate were performed once the cells had attached to the plate (4 h). Cells were then placed in cell culture incubation conditions. A total of 24 h later, 20 µL of MTT solution was added to each well and incubated for 3 h. Medium was removed, and precipitates solubilised in 100 µL DMSO. Absorbance was then measured at 570 nm using a Clariostar Plus microplate reader (BMG Labtech, Offenburg, Germany).

#### 4.4.3. Trypan Blue Exclusion Assay

Trypan blue exclusion assay was performed on MDA-MB-231 and MCF10A cells following treatment with DOX or DMSO. Following a published protocol [66], trypsinised cells were mixed with 0.4% Trypan blue solution and counted to determine the number of unstained (viable) and stained (nonviable) cells.

### 4.5. Data Analysis

Figures were arranged and statistical analyses were performed with GraphPad (Prism). Specific statistical analysis can be seen in figure legends. ANOVA with Bonferroni or Tukey post hoc analysis was performed for each data set to determine statistical significance.

### 4.6. Ethical Approval

Approval for all animal procedures was granted by the University of York Animal Welfare and Ethical Review Body. All procedures were carried out under authority of a UK Home Office Project Licence and associated Personal Licences.

## 5. Conclusions

Here, we have shown a new approach to VOC headspace sampling from cells in culture and mice. We present novel compound metabolisms not observed in cell lines or mice previously; notably, methyl halides and direct, quantified metabolic response due to drug treatment. We have demonstrated quantified fluxes (both consumption and production), in contrast to the measurement of presence versus absence [3,22,67].

Using this technique, we can identify cells from different tissues and whether cells from that tissue are cancerous or not. Furthermore, the response to cellular stress, from the chemotherapeutic Doxorubicin, is clearly defined in the volatile profile of both MDA-MB-231 breast carcinoma cells and noncancer MCF10A cells. However, the cancer cell line MDA-MB-231 revealed more significant alterations for MeCl, DMS, M57, 3-MP and n-hexane. This may have implications for monitoring chemotherapeutic treatments.

Our approach to investigating volatiles considers ambient environmental compounds and the processing of those compounds by the body. Ambient compounds which are taken up by cells or the body may be active metabolic substrates or accidentally metabolised; however, these reported metabolisms require further investigation. Volatile metabolisms in mammalian systems are an emerging field, and the processing of environmentally available VOCs takes into consideration the use of these compounds as potential substrates or chemical interactants.

The method presented aims to be translational to human breath. Longitudinal approaches may present an avenue to overcome confounding and conflicting results between the breath of individuals with similar pathologies. Organic systems have not evolved independently of environmental volatiles and their processing of them may be effective tools in biomarker discovery.

Using this approach may allow researchers to investigate volatile compounds in a new way for volatile biomarker discovery and diagnostic procedures. The compounds investigated here, including methyl halides, present an opportunity to explore metabolisms as they are processed by cells and present in cellular headspace and breath. Methyl chloride is consistently enhanced in mammalian breath and the cellular headspace, and its significant alterations in response to cellular stress may translate well into breath. Several compounds presented here show similar promise for human diagnosis, and further research is required to refine and describe the representative conditions that create specific metabolic outcomes.

## Figures and Tables

**Figure 1 metabolites-12-00599-f001:**
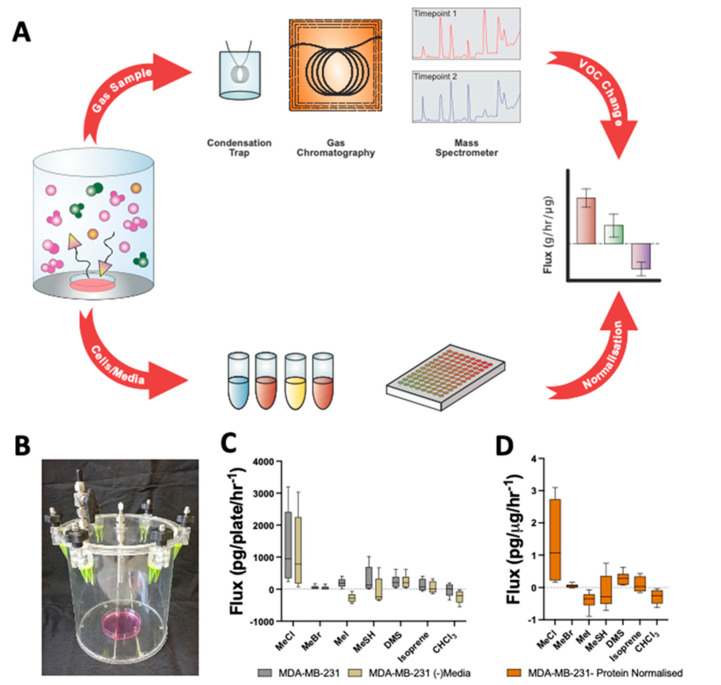
Direct volatile sampling of cellular headspace. (**A**) Schematic overview for methodological approach; headspace sampling and generation of VOC flux. (**B**) Image of collection chamber. (**C**) Selected volatile fluxes (g/h/plate) for 10 cm dishes containing DMEM media control only vs. plate containing MDA-MB-231 (mean ± SEM; *n* = 6). (**D**) Media subtracted and protein normalised VOC flux for MDA-MB-231 cells (mean ± SEM; *n* = 6). ANOVA followed by Bonferroni post hoc test was performed.

**Figure 2 metabolites-12-00599-f002:**
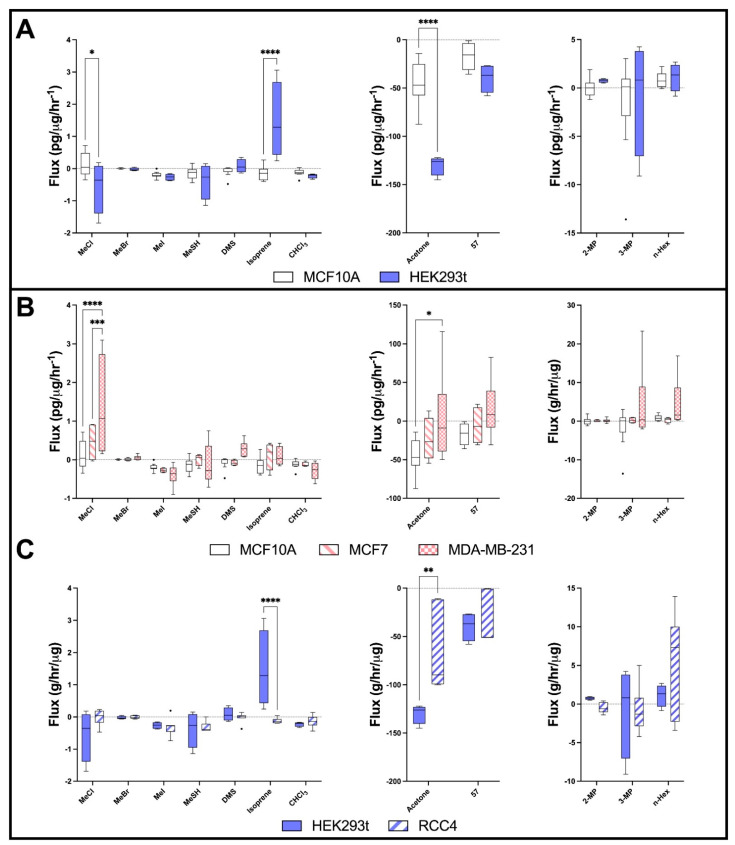
Cellular volatile profiles of breast- and kidney-derived cell lines. (**A**) Volatile flux (g/hr/µg) for noncancerous-derived cell lines, from breast; MCF10a and kidney; HEK293t. (**B**) Volatile flux for cancerous-breast-derived cell lines, MCF7 and MDA-MB-231. (**C**) Volatile flux for cancerous-kidney-derived cell line RCC4. Media subtracted and protein-normalised VOC flux for MCF10a (*n* = 9); MCF7 (*n* = 4); MDA-MB-231 cells (*n* = 6). CHCI3 = Chloroform, DMS = Dimethyl sulphide, MeBr = Methyl bromide, MeCl = Methyl Chloride, MeI = Methyl iodide, MeSH = Methanoethiol. Boxplot whiskers show median ± Tukey distribution. ANOVA followed by Bonferroni post hoc test was performed; * *p*  <  0.05; ** *p*  <  0.01; *** *p*  <  0.001; **** *p*  < 0.0001.

**Figure 3 metabolites-12-00599-f003:**
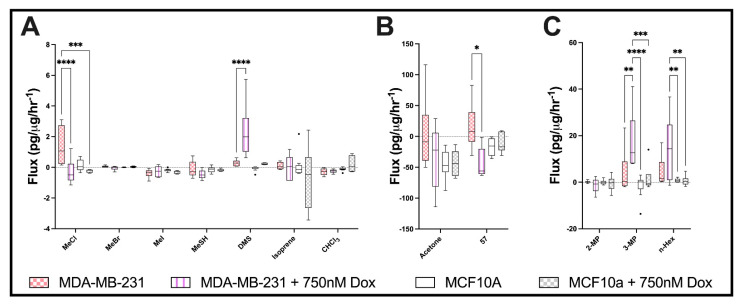
Doxorubicin induces volatile response in breast cell lines. (**A**–**C**) Boxplot for select volatile organic compounds (median ± Tukey distribution; *n* = 6). ANOVA followed by Tukey post hoc test was performed; * *p*  <  0.05; ** *p*  <  0.01; *** *p*  <  0.001; **** *p*  < 0.0001. Doxorubicin has been abbreviated to Dox.

**Figure 4 metabolites-12-00599-f004:**
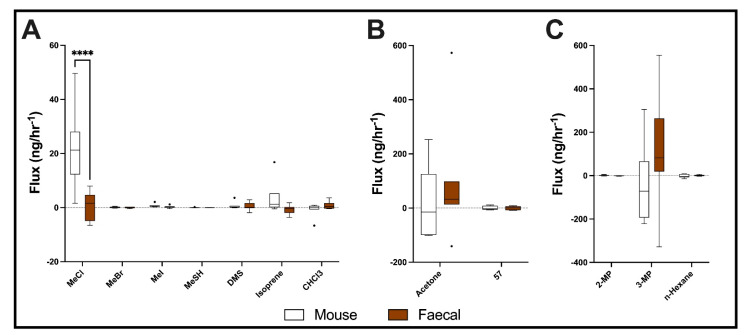
Volatile organic compounds from mouse breath and faecal material. (**A**–**C**) Boxplot for select volatile organic compounds from chambers with single mice vs. chambers with mice removed and faecal material. Flux in g/h (median ± Tukey distribution; *n* = 6 mice across 3 separate cages). ANOVA followed by Bonferroni hoc test was performed; **** *p*  <  0.0001.

## Data Availability

The data presented in this study are available in article or Appendix A.

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
