# Peer review of "Sampling and Analysis of Low-Molecular-Weight Volatile Metabolites in Cellular Headspace and Mouse Breath"

_metabolites, 2022, doi:10.3390/metabo12070599_

Round 1

Reviewer 1 Report

The manuscript titled "Sampling and analysis of low molecular weight volatile metabolites in cellular headspace and mouse breath" is well presented with a good novelty and significance. The author showed VOC headspace sampling from cells in culture and mice in a new approach which can be very useful in volatile biomarker discovery and diagnostic procedures.  The author also identified cells from different tissues (breast and kidney) and if cells from that tissue are cancerous or not using this headspace sampling techniques which is intriguing. Also, investigations with chemotherapeutic doxorubicin treatment in breast derived cells have studied well, and this work might have benefit in monitoring chemotherapeutic treatments. In addition, the methods (ex:, GC-MS, and others), figures and experimental results are nicely designed, studied and presented. Furthermore, the writing of the manuscript is very scientific, neatly presented, and easily understandable.

Overall, the work has good potential in helping future VOC biomarker discovery and in human diagnostic purposes.

I would recommend this manuscript to publish without any further revisions.

Author Response

Response to Reviewer 1 Comments

Point 1: Overall, the work has good potential in helping future VOC biomarker discovery and in human diagnostic purposes.

Response 1: We thank reviewer 1 for their kind words and for recognising the potential impact upon the field, our results and experimental design.

Reviewer 2 Report

The manuscript presents an interesting research topic and is extremely relevant today but it is not quite easy to follow and minor writting correction  are needed.

  1. line 34: However, In ....must me corrected.
  2. line 59: g-1, s-1...in g-1, s-1
  3. line 113hr-1 with hr-1
  4. line 331, 363: CO2 ...in CO2
  5. some abbreviations of the substances should be explained for those who do not use them regularly, like DMEM, DMSO

Is this technique promising to translate into clinics for specific diagnosis?

Author Response

The authors would like to thank the reviewer for their helpful comments. We have addressed point 1-5 in the revision.

In response to the comment regarding translatability of this method to the clinic we have added a dedicated paragraph in the conclusion section and drawn more comparisons to human studies throughout the revised document.

Reviewer 3 Report

The manuscript requires major revisions largely because the parts are not presented in correct logical and required order. Section 4 (Materials and Methods) should come before Section 2 (Results) according to journal convention.

Introduction

Lines 90-103 are statements more appropriate in the Discussion section. These lines should be replaced with a short paragraph indicating specific study objectives, identifying what aspects of volatile emissions from human cell lines and mice sample sources being studied and what results are expected (operating hypothesis).

Figure 1 and Lines 112-114 should be in the Materials and Methods

The Introduction avoids many references of VOC studies conducted on human and pathogen cell lines, particularly in the biomedical industry relating to diseases caused by microbes and metabolic disorders. The premise of the paper as well as the Conclusions suggest that this approach of investing VOC in human cell lines is a novel technique and idea. This is not the case and this area of research has been studied for many years. Both the Introduction and Conclusions should reflect this fact.

Some section subheading titles need to be changed to more accurately indicate what these sections are about (as follows):

2.1 Volatile emission rates from cell lines

Revise phraseology to volatile emissions (flux) rates in text descriptions

2.2 Volatile profiles by cell types

2.3 Chemical treatment effects on volatile emissions

2.4 Breath and fecal volatiles from mice

Discussion

Compare results to many published studies that have used analysis of human and other cell lines to do pathology diagnostics based on VOC emissions.

Conclusions

The approach developed in this study is not new and must be rephrased to only indicate what is specifically new (relative to the literature on cell line VOC analyses) which have not been included in this paper in the Introduction. 

References

Punctuation is missing in all references: Periods after author initials, semicolons after each author, periods after journal abbreviation, some volume numbers also missing. Appropriate spaces are missing after semicolons etc. Please check each reference for proper formatting for MDPI journals.

Author Response

Response to Reviewer 3 Comments

The authors would like to thank the reviewer for their in-depth review of the article. Their comments, insight and expertise have allowed us to improve the manuscript and we are grateful for their time and energy.

The first two of Reviewer 3’s comments are editorial matters. We note these comments and address the remaining comments after this brief acknowledgement and comment.

Point 1:“The manuscript requires major revisions largely because the parts are not presented in correct logical and required order. Section 4 (Materials and Methods) should come before Section 2 (Results) according to journal convention.”

“Figure 1 and Lines 112-114 should be in the Materials and Methods”

While we appreciate the reviewer’s approach we have kept figure 1 in the main text as this keeps the narrative, highlighting the method first. Because this is a narrative/stylistic change from the journals recommendation we refer to the editor’s decision. Figure 1 and lines 112-114 have therefore been kept in the results section for continuity.

We do not feel that we are authorized to make the requested revisions since this is not how the MDPI template arranges the document. We have contacted the editors for their advice on this.

Introduction:

Point 2: “Lines 90-103 are statements more appropriate in the Discussion section. These lines should be replaced with a short paragraph indicating specific study objectives, identifying what aspects of volatile emissions from human cell lines and mice sample sources being studied and what results are expected (operating hypothesis).”

We have followed the reviewer’s comments here, moved the section to the discussion and placed a short paragraph at the end of the introduction.

“This study aims to characterize volatile metabolisms of a select suite of volatiles in different cell types and status (disease). We hypothesis that volatiles collected from the headspace above these cells will differ significantly. Furthermore, treatment with chemotherapeutic agent, Doxorubicin will also produce significant, detectable alterations in the volatiles metabolized by cells. These volatile metabolisms, linked to phenotype and pathophysiology, could provide potential targets for diagnostic research. To test translatability of the method and select volatiles, mice breath will also be investigated.”

Point 3: “The premise of the paper as well as the Conclusions suggest that this approach of investing VOC in human cell lines is a novel technique and idea. This is not the case and this area of research has been studied for many years. Both the Introduction and Conclusions should reflect this fact.”

We thank the reviewer for highlighting confusion around our narrative. To address this, detail the published work available and to identify the specific novelty of the method we have added the following in the introduction:

“The majority of studies investigate volatile emissions (by definition, the production or dis-charge of a compound from an organism or system), generally through comparison of single time-point sample headspace peak area/concentrations relative to a background or control. These studies identify compounds that are produced by an organism/cell, but they do not quantify the rate at which the metabolism occurs. Uptake by human cells has been observed [22,23] however, compound consumption metabolisms are neither quantified nor understood, and longitudinal studies of cellular/organism metabolisms is a developing area of research. Here, we present a novel approach utilizing multiple time points to allow quantification of volatile compound metabolism (where metabolism, or flux, incorporates the potential for either emission or consumption) and allows longitudinal approaches which can identify and quantify changes in active cellular processing of volatiles.”

We have also made substantive efforts to include and acknowledge the previous work on cellular volatiles, and to point out the differences between the previous work and our method and approach. If the reviewer has any further article(s) that we have missed and they feel are critical to incorporate, we would appreciate if they could please identify them.

Point 4: “Some section subheading titles need to be changed to more accurately indicate what these sections are about”

We thank the reviewer for suggesting alterations to headings in the results section and have taken them onboard. We have made the choice to not use the term ‘emissions’ in regards to volatile flux because it specifically only refers to the production or dispersal of a compound from a system/cell/organism and this is specifically something that our system addresses, and is one of the primary sources of methodological novelty. Furthermore, we have altered the suggestion slightly from the suggested ‘chemical treatment’ to chemotherapeutic agent for clarity.

2.1 Volatile flux in cellular headspace

2.2 Volatile profiles by cell type

2.3. Effect of chemotherapeutic agent, Doxorubicin, upon cellular  volatile profiles

2.4. Breath and faecal volatiles from mice

Discussion:

Point 5: “Compare results to many published studies that have used analysis of human and other cell lines to do pathology diagnostics based on VOC emissions.”

We have attempted to contextualise the presence of the select volatiles in appropriate models and breath, however many of the presented compounds have not previously been reported. We have added extra content to draw comparisons to cellular models where appropriate.

“MCF-7 cells have been shown to release alkanes in response to oxidative stress [9] which is supported by the release of 3-MP and hexane in response to Doxorubicin, which has also been shown to induce oxidative stress [55,56].”

Conclusions:

Point 6: “The approach developed in this study is not new and must be rephrased to only indicate what is specifically new (relative to the literature on cell line VOC analyses) which have not been included in this paper in the Introduction.”

The authors thank the reviewer for pointing out that we require more clarity around novelty. We have added the following in the conclusion

“Here, we have shown a new approach to VOC headspace sampling from cells in culture and mice. We present novel compound metabolisms, not observed in cell lines or mice previously, notably, methyl halides and direct, quantified metabolic response due to drug treatment. We have demonstrated quantified fluxes (both consumption and production), in contrast to measurement of presence versus absence [3,22,55]. “

References:

Point 7: “Punctuation is missing in all references: Periods after author initials, semicolons after each author, periods after journal abbreviation, some volume numbers also missing. Appropriate spaces are missing after semicolons etc. Please check each reference for proper formatting for MDPI journals.”

We have addressed all points.

Round 2

Reviewer 3 Report

The authors have added adequate new references to cover literature involving analysis of VOCs in human cell cultures, etc. associated with human pathogens and disease. The manuscript still has some very significant structural abnormalities with incorrect order of sections according to science reporting convention of the journal. The Materials & Methods should follow the Introduction. The Conclusions should be elevated to its own section after the Discussion section, not a subsection of the Discussion. The Discussion section should provide more comparisons of results (of the current work) to those of other literature published work (using similar or different methods).  

Author Response

Response to reviewer 3 comments

We thank the reviewer for reviewing our submission further and for helping us to improve it.

Point 1:

“The manuscript still has some very significant structural abnormalities with incorrect order of sections according to science reporting convention of the journal. The Materials & Methods should follow the Introduction. The Conclusions should be elevated to its own section after the Discussion section, not a subsection of the Discussion.”

In response the authors have contacted the editors and their response (06/06/2022) is as follows: “We kindly ask you to keep the section order accepted by our guidelines (Material and Methods should be section 4)”. This is the standard convention of the metabolites journal which we have followed. We are therefore unable to alter the layout of the manuscript any further, but we thank the reviewer for their careful consideration of the narrative.

Point 2:

“The Discussion section should provide more comparisons of results (of the current work) to those of other literature published work (using similar or different methods).”

We thank the reviewer for further contextualisation of the results past those made in the previous revision. This has again helped to improve the document. We have gone through the compounds to add information on headspace volatiles linked to those investigated.

Line 256 “Of these 12 VOCs, this is the first report of methyl halide and chloroform metabolism in human cells and the breath of mice.”

Line 283 “]. HepG2 (hepatocarcinoma cells) and TBE (tracheobronchial epithelial cells) have been shown to produce DMS [23,45], whereas we have only shown production in MDA-MB-231 cells and in MCF10a and MDA-MB-231 cells following treatment with Doxorubicin.

Line 299 “Isoprene has been reported to be metabolized by HepG2 cells [23]. Here we have shown clear isoprene production by HEK293t and uptake by MCF10a and RCC4.”

Line 304 “2-MP has been shown to be produced by the lung cancer cell line NCI-H2807 [53] whereas we have only shown production by HEK293t cells. 3-MP uptake has been demonstrated in the lung cancer cell line A549 [54,55]. We have only shown significant production in response to Doxorubicin treatment in MDA-MB-231 cells.”

Line 317 “Uptake of acetone has been shown in the headspace of A549 and TBE cells [45,54] but emissions have been shown by VGP (vertical growth phase melanoma cells) [57] and A549 cells [26]. We have not shown consistent acetone production in any cell lines here but varying levels of consumption across all cells. HEK293t cells consumed the most acetone and cancerous cells showed relatively less consumption against non-cancerous cells.”

If the reviewer has any other specific literature we can reference that is within the scope of the work and we will include it. Again, we thank them for their time and energy in reviewing our manuscript.

Round 3

Reviewer 3 Report

The authors have made significant improvements in the manuscript and the formatting of reported sections do meet the requirements of Metabolites although it seems that the Conclusions could be elevated and reported as a separate major heading (for emphasis) instead of a subsection of the Discussion. Added comparative references in the Discussion are adequate and Reference formatting is much improved.

Author Response

Response to reviewer 3 comments

The authors would like to thank the reviewer for their continued efforts in reviewing our manuscript and for helping us in developing the work.

Point 1:

“Conclusions could be elevated and reported as a separate major heading (for emphasis) instead of a subsection of the Discussion”

Thank you for spotting this, the conclusions are now presented as section 5, following the guidelines from the journal.